# A scale-invariant log-normal droplet size distribution below the critical concentration for protein phase separation

**Tommaso Amico[1], Samuel Toluwanimi Dada[2], Andrea Lazzari[1], Michaela Brezinova[2], Antonio Trovato[1,3], Michele Vendruscolo[2]*, Monika Fuxreiter[1,4]*, Amos Maritan[1,3]***

[1]Department of Physics and Astronomy, University of Padova, Padova, Italy; [2]Centre for Misfolding Diseases, Department of Chemistry, University of Cambridge, Cambridge, United Kingdom; [3]National Institute for Nuclear Physics (INFN), Padova Section, Padova, Italy; [4]Department of Biomedical Sciences, University of Padova, Padova, Italy

**\*For correspondence:**
mv245@cam.ac.uk (MV);
monika.fuxreiter@unipd.it (MF);
amos.maritan@unipd.it (AM)

**Competing interest:** The authors declare that no competing interests exist.

## eLife assessment

In this **valuable** study, the authors analyze droplet size distributions of multiple protein condensates and their fit to a scaling ansatz, highlighting that they exhibit features of first- and second-order phase transitions. The experimental evidence is **solid**, and it prompts further research into the nature of the link between percolation and phase separation.

**Abstract** Many proteins have been recently shown to undergo a process of phase separation that leads to the formation of biomolecular condensates. Intriguingly, it has been observed that some of these proteins form dense droplets of sizeable dimensions already below the critical concentration, which is the concentration at which phase separation occurs. To understand this phenomenon, which is not readily compatible with classical nucleation theory, we investigated the properties of the droplet size distributions as a function of protein concentration. We found that these distributions can be described by a scale-invariant log-normal function with an average that increases progressively as the concentration approaches the critical concentration from below. The results of this scaling analysis suggest the existence of a universal behaviour independent of the sequences and structures of the proteins undergoing phase separation. While we refrain from proposing a theoretical model here, we suggest that any model of protein phase separation should predict the scaling exponents that we reported here from the fitting of experimental measurements of droplet size distributions. Furthermore, based on these observations, we show that it is possible to use the scale invariance to estimate the critical concentration for protein phase separation.

## Introduction

Many proteins have been shown to undergo a phase separation process into a liquid-like condensed state (*Brangwynne et al., 2009*; *Hyman et al., 2014*; *Banani et al., 2017*; *Alberti and Hyman, 2021*; *Lyon et al., 2021*). This process appears to be of physiological significance since it may lead to the formation of biomolecular condensates (*Brangwynne et al., 2009*; *Hyman et al., 2014*; *Banani*

*et al., 2017*; *Alberti and Hyman, 2021*; *Lyon et al., 2021*; *Buchan and Parker, 2009*; *Andersen et al., 2005*). As a consequence, it is closely controlled by the protein homeostasis system (*Mateju et al., 2017*; *Qamar et al., 2018*), and its dysregulation has been associated with a broad range of human diseases (*Mitrea et al., 2022*; *Vendruscolo and Fuxreiter, 2022*). It is therefore important to understand the fundamental nature of this process (*Lyon et al., 2021*; *Fuxreiter and Vendruscolo, 2021*; *Jawerth et al., 2020*) to provide insights for the identification of ways of modulating it through pharmacological interventions.

To help understand the nature of the transition underlying protein phase separation on the basis of recent experimental observations (*Stender et al., 2021*; *Kar et al., 2022*), here we study the distribution of the size of the droplets below the value of the concentration at which the transition occurs, which here is referred to as the critical concentration, $\rho_c$. This question appears as a promising starting point to develop new insights since it has been reported that proteins can form droplets of sizeable dimensions already well below the concentration at which phase separation occurs (*Stender et al., 2021*; *Kar et al., 2022*). This behaviour is not predicted by classical nucleation theory (*Li et al., 2012*), and not readily consistent with the idea that the protein phase separation process can be described as a first-order phase transition. This is because, in a first-order phase transition, nucleation takes place in a supersaturated system (*Shimobayashi et al., 2021*), while in a subsaturated system particles can still self-assemble, but with a probability that decreases exponentially with the size of the assemblies.

In the present study, we asked whether the experimental data on the droplet size distributions obey scale invariance, a general characteristic of complex systems, which is often used to reveal universal patterns underlying self-assembly. We report the observation that the droplet size distributions of the proteins FUS and α-synuclein follow a scale-invariant log-normal behaviour. These findings are consistent with a universal behaviour resulting from the presence of an increasingly large correlation length, $\xi$, as the concentration approaches the critical concentration from below. The correlation length is an emergent characteristic, and it is related to the typical spatial range over which density fluctuations are correlated. When $\xi$ is sufficiently large, one can expect scale invariance and finite-size scaling (*Brankov et al., 2000*) to occur within a range of lengths spanning from the molecular size to $\xi$. This means that physical observables cease to depend explicitly on numerous microscopic details that are peculiar to spatial scales smaller than $\xi$, thus leading to a universal behaviour characterized by quantities obtained by coarse graining over scales smaller than $\xi$.

At a first-order phase transition, $\xi$ is finite, and if it is not large enough, the length range discussed above remains too short to observe scale invariance. However, the vicinity of the spinodal line, where nucleation disappears as the dilute phase becomes unstable, to the coexistence curve, where nucleation appears as the dilute phase becomes metastable, might cause a large increase of $\xi$ as the first-order phase transition is approached by increasing the concentration. If it is not preceded by a first-order phase transition, the spinodal line would correspond to a second-order phase transition, resulting in infinite $\xi$, with scale invariance holding on all length scales larger than the molecular scale. Various alternative scenarios contributing to the emergence of significant correlations will be mentioned in the Discussion section. Nevertheless, our scaling analysis exhibits a high degree of generality, devoid of reliance on specific underlying models.

As an application of the above observations, we address the question of whether scale invariance holds for droplet size distributions near the coexistence curve. As a practical consequence, we use this observation to propose a procedure to overcome the challenge of estimating the critical concentration, $\rho_c$, which enters explicitly into our scaling analysis. Such challenge arises from the fact that close to the critical concentration, the timescale required for the equilibration of a system grows together with $\xi$, thus exceeding the timescale amenable to experimental observation.

Our analysis of experimental data indicates that: (i) scale invariance does indeed hold near the coexistence curve and (ii) the droplet size distribution is log-normal. Based on the properties of the scale-invariant log-normal distribution of droplet sizes, we investigate a correlation between the moments of the distribution and the distance from $\rho_c$.

Finally, we note that methods to assess the critical concentration are crucial for understanding the location of proteins in their phase diagram, their proximity to the phase boundary between the native and droplet states, and how pharmacological interventions can modify their phase behaviour. To address this problem, we report how the moments of the distribution can be used as a scale-invariant gauge to estimate the critical concentration. In this way, an accurate estimate of the critical

concentration is possible because it is based on measurements carried out away from the critical point, under conditions such that fluctuations are small, and hence experimental errors are smaller than in the proximity of the transition.

## Results

### Formulation of the scaling ansatz

Empirical evidence indicates that protein self-assembly into liquid-like condensates is characterized by: (i) a phase separation transition at a concentration $\rho_c$, (ii) a formation of droplets of sizeable dimensions already below $\rho_c$, and (iii) a droplet size distribution that, after an initial transient, does not change with the experimental observation time, although individual droplets can form, grow, shrink, and dissolve. An initial analysis of droplet size distributions observed experimentally led us to ask whether the region near the transition could be described in terms of a scaling theory, as commonly done for critical phenomena (**Shimobayashi et al., 2021**), as summarized above. We also note that this approach is analogous to analysing the cluster size distribution in a percolation problem (**Stauffer and Aharony, 2018**).

In our analysis, we called $P > (s|\rho)$ the survival distribution function (SDF = 1 - CDF, where CDF is the cumulative distribution function) corresponding to the probability to observe a droplet of size greater than $s$, when the concentration is $\rho$. The distance from the critical concentration $\rho_c$ is measured in terms of the dimensionless variable $\tilde{\rho} = (\rho - \rho_c)/\rho_c$, which also allows to compare data from different experiments, as explained below. $P > (s|\rho)$ in general depends separately on $s$, $\rho$, and many other parameters characterizing the process, including temperature. However, if scale invariance holds in the vicinity of $\rho_c$, that is when $|\tilde{\rho}|$ is small and the correlation length of the system is large enough, we would expect $P > (s|\rho)$ to depend on $\rho$ and on other details pertaining to the microscopic scales only through the characteristic droplet size, $s_c$. The characteristic size $s_c$ is defined, apart from a proportionality constant (see below), as the ratio of the second to the first moment of the droplet size distribution. This leads us to formulate the following scaling ansatz (**Stauffer and Aharony, 2018**).

$$P_> \left(s|\rho\right) = s^{-\alpha} f\left(\frac{s}{s_c}\right) \tag{1}$$

where $s_c$ depends on $\rho$, and it is expected to diverge at the critical concentration as (**Brankov et al., 2000**; **Stauffer and Aharony, 2018**)

$$s_c = a \left|\frac{\rho - \rho_c}{\rho_c}\right|^{-\varphi} = a \left|\tilde{\rho}\right|^{-\varphi}, \tag{2}$$

where $\alpha \geq 0$ and $\varphi > 0$ are critical exponents, $a$ is a constant and $f$ is the so-called scaling function (**Brankov et al., 2000**).

Thus, the scaling of the SDF is equivalent to saying that, apart from the singular behaviour $s^{-\alpha}$, the remaining $s$ and $\rho$ dependence occurs only through the ratio $s/s_c$. All extra dependencies are encapsulated in $s_c$ through the constant $a$, $\rho_c$ and, possibly, on the specific form of the scaling function $f$. A consequence of scaling and singular behaviour described by **Equation 2**, that is the divergence of the characteristic size of the droplets, is that details of any specific system may not affect the value of the critical exponents, which are therefore expected to be universal, that is independent of specific details of the system.

To determine the exponents $\alpha$ and $\varphi$, we introduce the moments of $P_> \left(s | \rho\right)$ for $k > 0$:

$$\left\langle s^k \right\rangle = \int_0^\infty s^k \cdot \left(-\frac{dP_>(s | \rho)}{ds}\right) ds = c_k \cdot s_c^{k-\alpha} \tag{3}$$

where the scaling ansatz **Equation 1** has been used in the last step and $c_k$ is given by

$$c_k = k \int_0^\infty x^{k-1-\alpha} f\left(x\right) dx$$

which depends on the function $f$ but is independent of $\tilde{\rho}$. From **Equation 3**, we deduce that

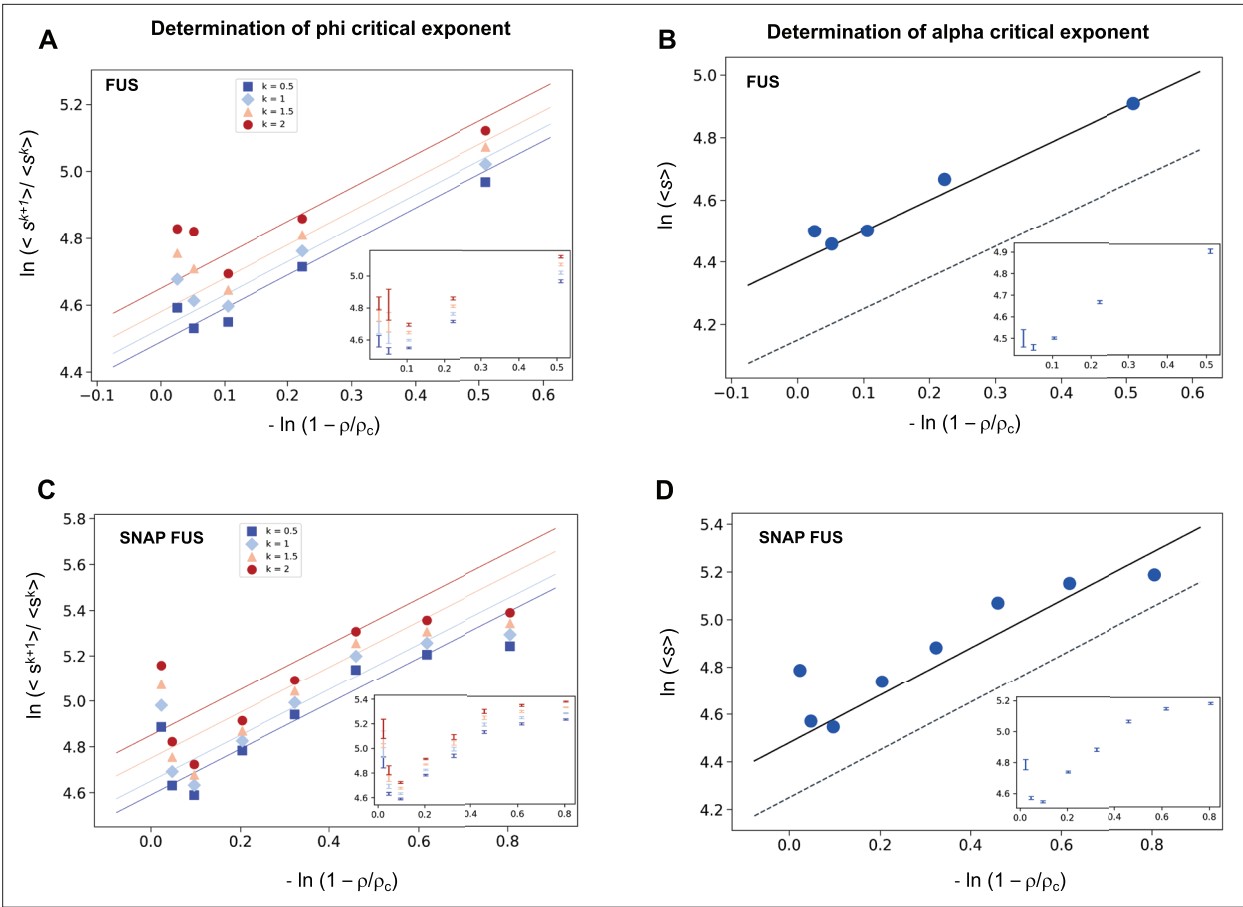

**Figure 1.** Determination of the critical exponents for FUS of the scaling invariance. Determination of the exponent $\varphi$ for FUS (**A**) and SNAP-tagged FUS (**C**). The ratios of the average moments of the droplet sizes ($<s^{k+1}>/<s^k>$ , at $k$ = 0.5, 1, 1.5, 2, **Equation 4**) are represented at various distances from the critical concentration ($|\tilde{\rho}|$). The exponent $\varphi$ for each value of $k$ was determined by error-weighted linear regressions. The exponent $\varphi$ and its error were determined as mean and standard deviation of the three independent measurements (**Equations 15 and 16**). Error bars are shown in inset for graphical clarity. Determination of the exponent $\alpha$ for FUS (**B**) and SNAP-tagged FUS (**D**). The mean of the droplet size distributions is plotted at various distances from the critical concentration ($|\tilde{\rho}|$). The value of the exponent $m$ was determined by error-weighted linear regression (**Equation 6**), using $\varphi$ = 1, where the errors were standard deviations of the three independent measurements (**Equation 16**). Error bars, which were obtained as the standard deviation of the three independent measurements are shown in inset for graphical clarity. Error-weighted linear regressions are performed in both cases excluding the data point at the lowest concentration $\rho$ = 0.125 μM. The fit corresponding to the scaling ansatz, compatible with $\varphi$ = 1 and $\alpha$ = 0, is represented by a dashed grey line with a slope of 1.

$$\frac{\langle s^{k+1} \rangle}{\langle s^k \rangle} = \frac{c_{k+1}}{c_k} \cdot s_c = \frac{c_{k+1}}{c_k} \cdot a|\tilde{\rho}|^{-\varphi} \qquad (4)$$

If the scaling ansatz of **Equation 1** is correct, by plotting the ratio of moments $\langle s^{k+1} \rangle / \langle s^k \rangle$ for various values of $k$ as a function of $1/|\tilde{\rho}|$ in a double logarithmic scale, we should obtain straight parallel lines with slope $\varphi$ and intercept $\ln(ac_{k+1}/c_k)$.

## Scaling behaviour of the droplet size distributions of FUS

We investigated the validity the ansatz in **Equation 2** using experimental data on the RNA-binding protein FUS, which are available for both the untagged and the SNAP-tagged protein (**Kar et al., 2022**; **Supplementary file 1**). We calculated the SDF, **Equation 2**, and its moments, **Equation 3**. We then plotted the moment ratios versus the inverse distance from the critical concentration $1/|\tilde{\rho}|$ in double logarithmic scale (**Figure 1**). We used the estimates of the critical concentrations, $\rho_c$ = 5.0 μM for FUS and $\rho_c$ = 5.4 μM for SNAP-tagged FUS, obtained below, in a self-consistency check of the validity of the scaling ansatz. We observed that the moment ratios at different distances from the critical concentration fall onto straight lines, as predicted by the scaling ansatz. In addition, the weighted

average slope (see *Equations 15 and 16* below) of the lines for different moment ratios is 0.95 ± 0.05 for untagged FUS, and 0.95 ± 0.05 for SNAP-tagged FUS, which is in good agreement with $\varphi = 1$ for the exponent in the scaling ansatz in *Equation 1* (*Figure 1A, C*).

Having determined the exponent $\varphi$, we can also determine the exponent $\alpha$ using *Equation 3*

$$\langle s^k \rangle = c_k \cdot s_c^{k-\alpha} \propto \left| \frac{\rho - \rho_c}{\rho_c} \right|^{-\varphi \cdot (k-\alpha)} \tag{5}$$

The exponent $\alpha$ is then calculated from

$$m = (k - \alpha)\,\varphi \rightarrow \alpha = k - \frac{m}{\varphi} \tag{6}$$

where $m$ is the slope of the linear fit of the double logarithmic plot of <s> (the first moment, $k = 1$) versus $1/|\tilde{\rho}|$. We then plotted the mean droplet size, <s>, versus the inverse distance from the critical concentration $1/|\tilde{\rho}|$ on a natural logarithm scale, which could be fitted using a line with a slope $m = 0.99 \pm 0.05$ for untagged FUS, and $0.93 \pm 0.07$ for SNAP-tagged FUS (*Figure 1B, D*), which is consistent with $m = 1$. Using the value of $\varphi = 1$, determined based on *Equation 4*, we obtain $\alpha = 0$. Taken together, these data support the validity of the scaling ansatz of *Equation 1*.

## The droplet size distribution of FUS is log-normal

The above analysis suggests that the droplet size distribution may follow a log-normal distribution. The scaling ansatz of *Equation 1* for the SDF is equivalent to the following scaling for the probability density distribution

$$P(s \mid \rho) = \frac{-dP_>(s \mid \rho)}{ds} = s^{-\alpha - 1} \cdot F(s/s_c), \quad s_c \propto |\tilde{\rho}|^{-\varphi} \tag{7}$$

where $f$, the scaling function in *Equation 1*, and $F$ are related as follows

$$f(z) = z^\alpha \int_z^\infty x^{-\alpha - 1} F(x)\, dx \tag{8}$$

The log-normal droplet size distribution $P(s|\rho)$ is

$$P(s \mid \rho) = s^{-1} \frac{1}{\sigma \sqrt{2\pi}} \exp \left\{ -\frac{(\ln(s/s_0))^2}{2\sigma^2} \right\} \tag{9a}$$

with

$$s_0 \equiv s_c e^{-3\sigma^2/2}, \text{where } s_c = \frac{\langle s^2 \rangle}{\langle s \rangle} \tag{9b}$$

being the characteristic droplet size as defined above. Consequently, the size SDF is

$$P_>(s|\rho) = \frac{1}{2} \cdot erfc \left( \frac{\ln(s/s_0)}{\sigma \cdot \sqrt{2}} \right) \tag{10}$$

The values of $s_0$ and $\sigma$ can be determined as the average and the variance, of $\ln(s/u)$ obtained at each concentration:

$$\ln(s_0/u) \equiv \langle \ln(s/u) \rangle$$
$$\sigma^2 \equiv Var\left( \ln(s/u) \right) \equiv \langle \ln^2(s/u) \rangle - \langle \ln(s/u) \rangle^2 \tag{11}$$

where $u$ is an arbitrary (and irrelevant) constant with the same units as $s$. In the following, when not stated, it is implicitly assumed that $u = 1$ in the same units as $s$. The droplet sizes follow a log-normal distribution only if the SDFs or, equivalently, the size distribution functions, multiplied by $s$, collapse when plotted versus *Equation 9a* or *Equation 10* with the values of $s_0$ and $\sigma$ of each droplet size distributions obtained at different concentrations (*Equation 11*). We determined $s_0$ and $\sigma$ values for each distribution (*Figure 2A, B*) and plotted the properly rescaled size distribution functions versus

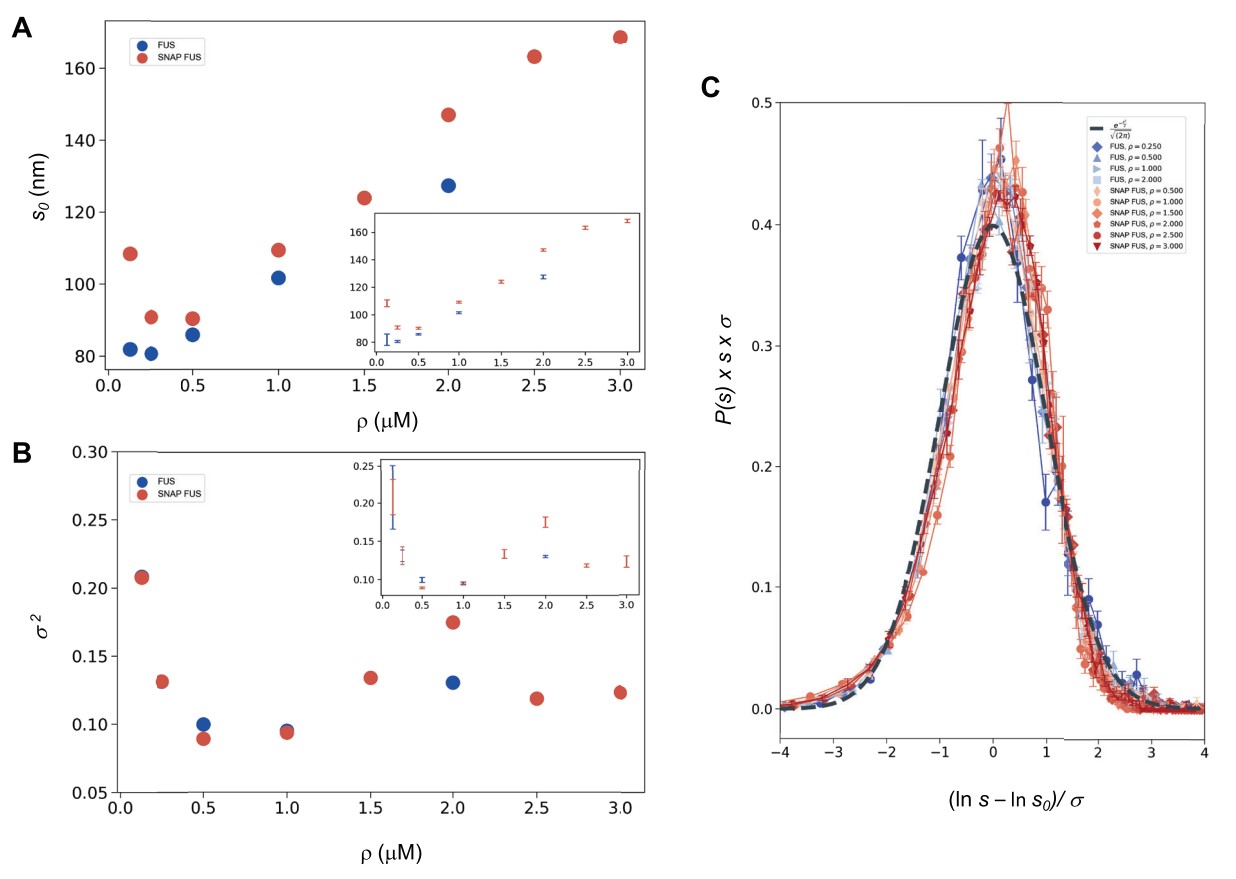

**Figure 2.** Log-normal behaviour of FUS and SNAP-tagged FUS size distributions below the critical concentration. Variation of the size distribution with protein concentration: ln $s_0$ (**A**) and $\sigma$ (**B**). ln$s_0$ and $\sigma$ (inset) were computed for FUS (blue) and SNAP-tagged FUS (red) using **Equation 11**. Error bars (in inset for graphical clarity) are estimated from three independent measurements. While droplet sizes increase with concentration, the width of distribution does not change considerably. (**C**) The collapse of the droplet size distribution functions is consistent with a log-normal behaviour. The droplet size distribution functions for both untagged FUS (blue) and SNAP-tagged FUS (red) are plotted after rescaling the sizes by the ln $s_0$ and $\sigma$ values, the first and second moments of the logarithm of the droplet size distribution, which are a function of the concentration. The rescaled curves for both the untagged and the tagged protein collapse to the normal distribution (grey dashed), as expected when the non-rescaled droplet sizes follow a log-normal distribution.

[ln($s/s_0$)]/$\sigma$ (**Figure 2C**). We observed that the size distribution functions collapsed for both FUS and SNAP-tagged FUS (**Figure 2C**). Furthermore, the collapsed curve overlapped with the analytic log-normal distribution we computed with $s_0 = 1$, $\sigma = 1$, the normal distribution in the rescaled variables.

The observed collapse supports the observation that droplet size distributions from different experiments follows a log-normal behaviour.

## Independence of the variance of the distribution from the concentration of FUS

The log-normal behaviour described above is consistent with the scale invariance underlying **Equation 1** if the variance of the log-normal distribution, $\sigma$, is independent of $\rho$ or, equivalently, of $\tilde{\rho}$ (**Giometto et al., 2013**). Indeed, comparing **Equation 1** with **Equation 7**, **Equation 9a**, **Equation 9b**; and **Equation 10** we obtain that the scaling invariance holds with

$$f(x) = \frac{1}{2} erfc \left[ \frac{\ln(xe^{3\sigma^2/2})}{\sigma\sqrt{2}} \right]$$

$$F(x) = \frac{1}{\sigma\sqrt{2\pi}} \exp\left\{ -\left[ \frac{\ln(xe^{3\sigma^2/2})}{\sigma\sqrt{2}} \right]^2 \right\}$$

(12)

with $\alpha = 0$ and $\varphi = 1$ (**Figure 1**). Notice that the log-normal distribution implies that $\alpha = 0$ whereas the value of $\varphi$ is not determined a priori. Furthermore, the $k$th moment of the log-normal distribution is

$$\langle s^k \rangle = s_c^k e^{\sigma^2 k(k-3)/2}, \tag{13}$$

which, compared with the scaling prediction **Equation 3**, is also consistent with $\alpha = 0$, appropriate for the log-normal, and

$$c_k e^{\sigma^2 k(k-3)/2} \tag{14}$$

which is independent of $\tilde{\rho}$ if $\sigma$ is independent of $\tilde{\rho}$, see **Equation 3**.

This prediction is verified in **Figure 2B**, where the $\sigma$ (**Hyman et al., 2014**) values are shown to be nearly uniform at different concentrations, with the exception of the data at the lowest concentration values, that is those furthest away from the critical concentration. These results are consistent with the scaling ansatz in the vicinity of $\tilde{\rho} = 0$.

Our analysis does not exclude the possibility that $\sigma$ might depend on specific experimental conditions, even though the data that we analysed are suggestive of at most a weak dependence. We note that such dependence, even if present, does not invalidate the scaling, as long as the exponents do not depend on the experimental conditions.

## Estimation of the critical concentration of FUS using the scale invariance

The fact that the scaling ansatz is satisfied for different set of experiments opens a possibility to estimate the critical concentration. The scaling with $\varphi = 1$ predicts $\left( \langle s_k \rangle \right)^{\frac{-1}{k}}$ versus $\rho$ to be a straight line with a slope depending on $k$. It is important to note that this is the consequence only of the scaling ansatz and not of the log-normal distribution. The line should intersect the $\rho$-axis at the critical concentration providing an estimate of $\rho_c$.

We illustrate this process using the FUS data by plotting $\left( \langle s_k \rangle \right)^{\frac{-1}{k}}$ versus $\rho$. As expected, for different values of $k$ we obtained a straight line fit of the points near $\rho_c$ (**Figure 3**). Due to experimental uncertainties, the various lines, one for each value of $k$, lead to a slightly different estimate of $\rho_c$. The average of the estimated $\rho_c$ values is 5.0 ± 0.2 μM for FUS and 5.4 ± 0.4 μM for SNAP-tagged FUS (**Figure 3**). The different $\rho_c$ values predicted based on the scaling ansatz using different $k$ values enable the estimation of error of the predicted critical concentration (**Figure 3**). Both estimates are higher than the values of $\rho_c$ originally reported ($\rho_c = 2$ μM of untagged and $\rho_c = 3$ μM of tagged FUS;

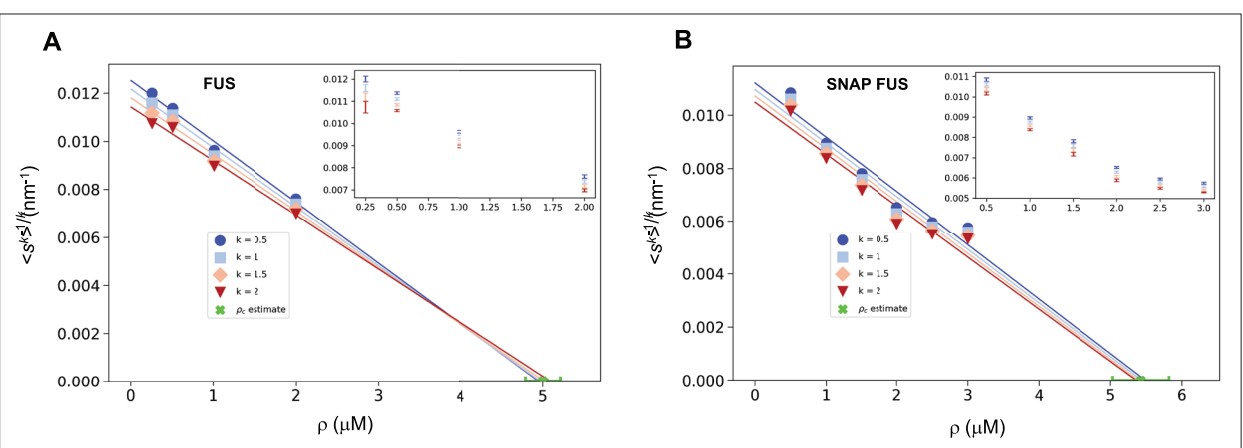

**Figure 3.** Estimation of the critical concentration of FUS using the scale invariance. Critical concentration of FUS (5.0 ± 0.2 μM) (**A**) and SNAP-tagged FUS (5.4 ± 0.4 μM) (**B**). The scaling model predicts that the function of the moments plotted versus the concentration $\rho$ becomes a straight line near the critical concentration $\rho_c$ and intersects the $\rho$-axis at $\rho_c$, independently of the value of $k$. Error-weighted linear regressions are performed in both cases excluding the data point at the lowest concentration $\rho = 0.125$ μM. The resulting estimate of the critical concentration is shown in green along with the corresponding standard deviation, estimated from three independent measurements.

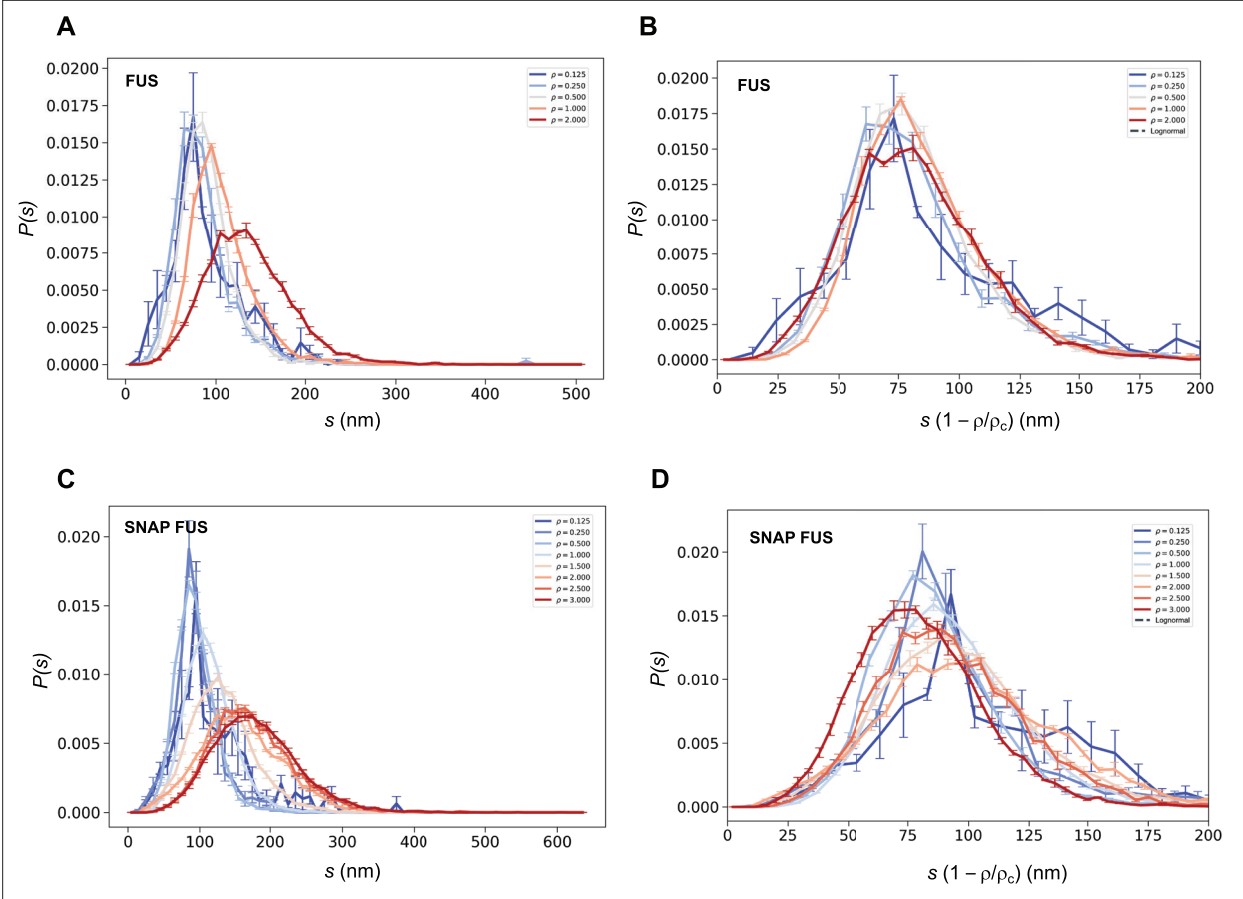

**Figure 4.** Collapse of the droplet size distributions of FUS as predicted by the scale invariance. If the scaling ansatz of *Equation 2* holds, the standard deviation $\sigma$ of the log-normal distribution should not depend on the distance from the critical concentration, and a collapse should be achieved by rescaling the size with the distance $|\tilde{\rho}|$ from the critical concentration. (**A, C**) Droplet size distributions derived from the experimental data of untagged FUS at 0.125, 0.25, 0.5, 1.0, and 2.0 µM concentrations (**A**) and SNAP-tagged FUS at 0.125, 0.25, 0.5, 1.0, 1.5, 2.0, 2.5, and 3.0 µM concentrations (**C**), and their standard error of the mean from three independent measurements (*Stender et al., 2021*). (**B, D**) Collapse of the droplet size distributions rescaled by the estimated critical concentration. The error bars show the standard error of the mean from the three independent measurements (*Stender et al., 2021*).

*Kar et al., 2022*) but consistent with them once taken into account the behaviour of the plateau of the absorbance of a spin-down assay used in that work. We also note that our estimates are compatible with other ones recently reported (*Patel et al., 2015*).

We then used both the values of $\rho_c$ to probe the collapse of the size distribution functions (*Figure 4*). We observed that both in cases of untagged and tagged FUS, the SDFs collapsed with our estimated value of $\rho_c$ (*Figure 4*).

These results indicate that the scaling model can be used to estimate the critical concentration based on the distribution of droplet sizes.

## Estimation of the critical concentration of α-synuclein using the scale invariance

As α-synuclein droplets (called clusters) were recently reported below the critical concentration (*Ray et al., 2023*), we aimed at characterizing whether such droplets also follow scale-invariant size distribution. Using A90C α-synuclein labelled with Alexa Fluor 647, we monitored droplet formation as previously described (*Dada et al., 2023*). We measured droplet sizes at 5% PEG concentration 10 min after detection of liquid-like condensates using increasing concentrations of α-synuclein (20, 40, 50, 60, 75, 80, and 100 µM) (*Supplementary file 2*). Using $k$ values of 0.25, 0.75, 1.25, 1.75, the critical exponents in the scaling ansatz (*Equations 4 and 6*), obtaining $\varphi = 1.3 \pm 0.2$ (*Figure 5A*) and $m =$

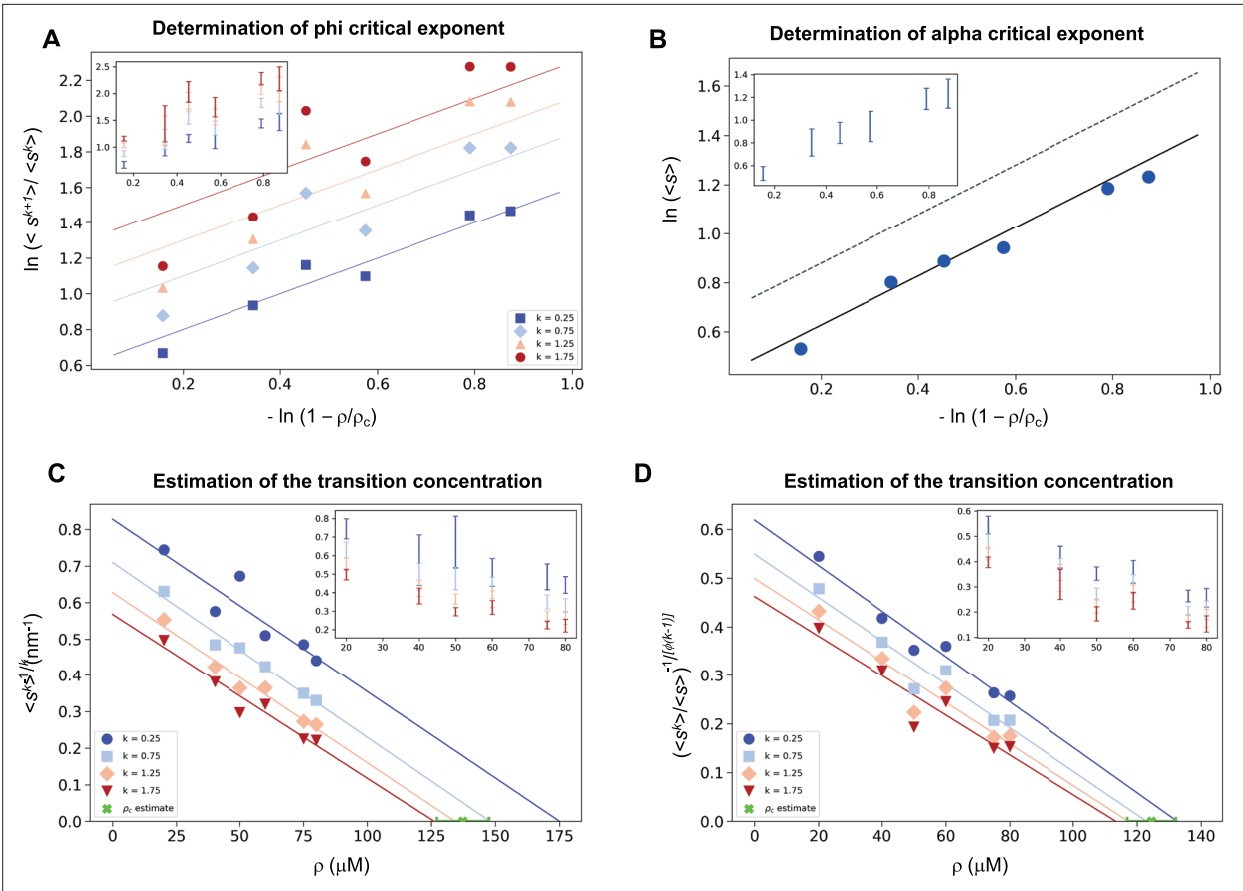

**Figure 5.** Estimation of the critical concentration of α-synuclein using the scale invariance. (**A**) Determination of the critical exponent $\varphi$. The ratios of the average moments of the droplet sizes ($<s^{k+1}>/<s^k>$, at $k$ = 0.25, 0.75, 1.25, 1.75, **Equation 4**) are represented at various distances $|\bar{\rho}|$ from the critical concentration. The exponent $\varphi$ and its error for each value of $k$ were determined as a mean and standard deviation of the three independent measurements (**Equations 15 and 16**). Error bars are shown in inset for clarity. (**B**) Determination of the critical exponent $m$. The mean of the droplet size distributions is plotted at various distances $|\bar{\rho}|$ from the critical concentration. The value of the exponent $\alpha$ was determined by error-weighted linear regression (**Equation 6**), using $\varphi$ = 1, where the errors were standard deviations of the five independent measurements (**Equation 16**). Error bars, which were obtained as the standard deviation of the five independent measurements are shown in inset for clarity. Error-weighted linear regressions were performed. The fit corresponding to the scaling ansatz, compatible with $\varphi$ = 1 and $\alpha$ = 0, is represented by a scattered grey line with a slope of 1. Determination of the critical concentration for α-synuclein using the scaling ansatz in two different ways, either using **Equation 18** (**C**), resulting in $\rho_c$ = 137 ± 10 μM, or **Equation 19** (**D**), resulting in $\rho_c$ = 125 ± 7 μM, which are consistent within errors. The scaling model predicts that the function of the moments plotted versus the concentration $\rho$ becomes a straight line near the critical concentration $\rho_c$ and intersects the $\rho$-axis at $\rho_c$, independently of the value of $k$. Error-weighted linear regressions were performed. The resulting estimates of the critical concentration are shown in green along with the corresponding standard deviation, estimated from four independent measurements. In panel D, $\varphi$ was constrained to 1.0 using **Equation 19**.

$1 - \alpha = 0.9 \pm 0.2$ (**Figure 5B**). As control, we also determined the critical exponents using the scaling ansatz in a different way, using **Equation 19**, obtaining $\varphi = 1.1 \pm 0.2$ and $m = 1 - \alpha = 0.8 \pm 0.2$, corroborating the validity of the scale-invariant model. We then estimated the critical concentration using the two methods obtaining $\rho_c$ = 137 ± 10 μM, using **Equation 18** (**Figure 5C**) and $\rho_c$ = 125 ± 7 μM, using **Equation 19** (**Figure 5D**). Images of the droplets at the different concentrations are shown in **Figure 6A**. The droplet size distributions are stationary below the critical concentration, as expected (**Figure 6B, C**; **Supplementary file 3**).

## Discussion

Although growing experimental evidence indicates the presence of protein condensates both in vitro and in vivo (**Brangwynne et al., 2009**; **Hyman et al., 2014**; **Banani et al., 2017**; **Alberti and Hyman, 2021**; **Lyon et al., 2021**; **Boija et al., 2018**), their exact nature, and the mechanisms responsible for their formation are not fully understood. For example, it is unclear whether the droplets observed

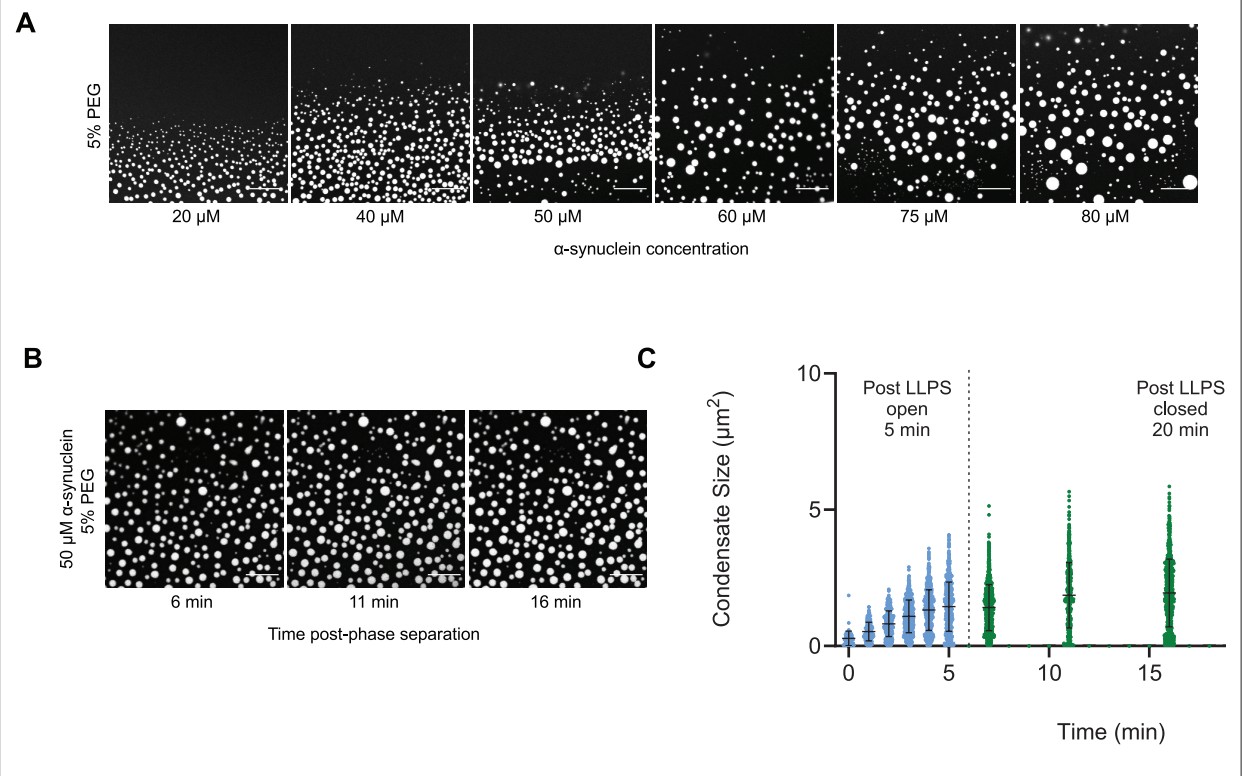

**Figure 6.** The droplet size distribution is stationary below the critical concentration. (**A**) Images of α-synuclein droplets at increasing concentrations of α-synuclein. (**B, C**) After an initial transient of 5 min, the droplet size distributions remain approximately stationary below the critical concentration, as shown for the case of 50 µM α-synuclein concentration. Results are reported for five replicates.

below the saturation concentration and those detected above result from the same process. Here, we aimed at describing phase separation using a scaling ansatz.

With this aim, we analysed a series of droplet size distributions of FUS and α-synuclein. We found that the droplet size distributions follow a scale-invariant log-normal behaviour. Scale invariance means that the description of the behaviour of a system remains the same regardless of the scale of observation. It has been seen for example in self-similar systems like fractals, which repeat patterns at different length scales (*Song et al., 2005*). Scale invariance is commonly observed in physics, biology, and economics, where it helps understand complex systems by identifying consistent patterns and fundamental properties (*Stanley, 1999*). The critical exponents of the scaling model are the same for different systems. We also note the generality of the log-normal model for nucleation and grain size growth ranging from crystal seeding (*Binder and Virnau, 2016*; *Bergmann and Bill, 2008*) to the mass size distributions in organism growth of various sea organisms (*Giometto et al., 2013*).

It is interesting to discuss the links of the scale invariance with the theoretical models that have been proposed to date to explain the phenomenon of protein phase separation. A commonly adopted framework is the Flory–Huggins theory of phase separation (*Flory, 1942*; *Huggins, 1942*; *Michaels et al., 2023*). In its simplest form, this theory describes a first-order phase separation in a system of homopolymers, which can also be adapted to polyampholytes (*Overbeek and Voorn, 1957*). The Flory–Huggins theory has been extended to associative polymers, with the aim of modelling the sequence composition of proteins, by Flory and Stockmayer, who described the phase separation in terms of a third-order gelation process (*Flory, 1941*; *Stockmayer, 1943*). Semenov and Rubinstein modified the Flory–Stockmayer theory, reporting that the gelation process is not a real phase transition, as all the derivatives of the free energy are analytical at the gelation point (*Semenov and Rubinstein, 1998*). It was also argued that another generalization of the Flory–Stockmayer theory (*Tavares et al., 2010*) could describe phase transitions in cytoskeletal networks (*Bueno et al., 2022*). In an older study, γ-crystallin was reported to undergo a second-order phase separation process (*Thomson et al., 1987*), consistently with observations in lysozyme solutions (*Ishimoto and Tanaka, 1977*).

More recently, it has been suggested that the protein phase separation process is coupled with percolation (*Mittag and Pappu, 2022*). It is notable that percolation models exhibit a continuous transition marked by a diverging correlation length and a scale-invariant droplet size distribution (*Stauffer and Aharony, 2018*), resembling the general form as presented in *Equations 1 and 2*. However, the critical exponents for 3D percolation at the critical threshold are $\alpha = 1.19$ and $\varphi = 2.21$, and in mean-field ($d > 6$) scenarios, they are $\alpha = 1.5$ and $\varphi = 2$, as per existing literature (*Stauffer and Aharony, 2018*). In contrast, our scaling analysis is consistent with $\alpha = 0$ and $\varphi = 1$. Therefore, a phase separation coupled with percolation model (*Mittag and Pappu, 2022*) could be consistent with the scaling behaviour reported here if this coupling would induce a shift in the universality class from that of standard percolation (*Stauffer and Aharony, 2018*). This possibility should be further investigated to determine whether this type of models could predict the critical exponents $\alpha = 0$ and $\varphi = 1$.

Scale invariance may appear to be in contrast with the Flory–Huggins theory, which characterizes droplet formation as a consequence of nucleation processes within a metastable state of a supersaturated system. In such cases, however, when the correlation length exceeds the cellular dimensions, the remnants of what should have been a first-order phase transition in an infinite system may effectively exhibit indistinguishable characteristics from a second-order phase transition. We also note that a power law distribution, but above the transition temperature, has recently been reported for nucleoli (*Lee et al., 2023*).

Quite generally, the existence of scale invariance for the droplet size distribution, at least under the conditions investigated, imposes stringent constraints on theoretical models that aim to elucidate protein phase separation. The importance of a scaling analysis lies in its ability to uncover the fundamental aspects of universal phenomena, transcending models confined solely to specific systems for which they were originally designed.

As a practical consequence of the scaling model, the moments of the droplet size distribution can be used to obtain an estimate of the critical concentration for phase separation.

## Conclusions and perspectives

It has been challenging to determine critical concentrations in the study of protein phase separation, particularly in cell systems. As a consequence, it remains still largely unclear when the observed protein droplets are formed at subsaturation concentrations, and when they represent phase-separated condensates at supersaturation concentrations. To make progress towards understanding this problem, we have analysed the probability distributions of the droplet sizes at different concentrations. The scale-invariant log-normal behaviour that we have reported offers a tool to determine the critical concentration. This approach should enable the positioning of a system with respect to the phase boundary when experimental data could be obtained about the length (1D), area (2D), or volume (3D) of the condensates. We anticipate that this method will find applications in the study of protein condensates in the cell.

We also note that the scale invariance of the droplet size distribution that we have reported is characteristic of critical phenomena, which tend to be highly sensitive to environmental conditions. The formation and dimensions of protein droplets could therefore be rather tunable. This feature would appear to be favourable for the control of the formation of biomolecular condensates by the protein homeostasis system, and also suggest that protein phase separation could be efficiently modulated pharmacologically.

## Methods
### Expression and purification of α-synuclein

Human wild-type α-synuclein and A90C cysteine variants were purified from *Escherichia coli* BL21 (DE3)-gold (Agilent Technologies) expressing plasmid pT7-7 encoding for α-synuclein as previously described (*Dada et al., 2023*; *Fusco et al., 2014*; *Hardenberg et al., 2021*). Following purification in 50 mM trisaminomethane-hydrochloride (Tris-HCl) at pH 7.4, α-synuclein was concentrated using Amicon Ultra-15 centrifugal filter units (Merck Millipore). The protein was subsequently labelled with a 1.5-fold molar excess of C5 maleimide-linked Alexa Fluor 647 (Invitrogen Life Technologies) overnight at 4°C with constant mixing. The excess dye was removed using an Amicon Ultra-15 centrifugal filter unit and used immediately for phase separation experiments.

### Determination of the droplet distribution of α-synuclein

The experimental conditions were determined from a previously described phase boundary (**Dada et al., 2023**). To induce liquid droplet/condensate formation, wild-type α-synuclein was mixed with an A90C variant labelled with Alexa 647 at a 100:1 molar ratio in 50 mM Tris-HCl and 5% polyethylene glycol 10,000 (PEG) (Thermo Fisher Scientific). The final mixture was pipetted onto a 35-mm glass-bottom dish (P35G-1.5-20-C; MatTek Life Sciences) and immediately imaged on Leica Stellaris Will inverted stage scanning confocal microscope using a 40×/1.3 HC PL Apo CS oil objective (Leica Microsystems) at room temperature. The excitation wavelength was 633 nm for all experiments. For liquid droplet size characterization, images were captured 10 min post-liquid droplet formation. To analyse the possible time dependence of the liquid droplet size distribution, phase separation was induced, and 5 min post onset of phase separation, the glass-bottom dish containing the experiment was sealed and closed to maintain a stable and controlled environment. Subsequently, images were captured at each designated time point. All images were processed and analysed in ImageJ (NIH). Images were analysed by applying a threshold function in ImageJ that excluded the background of the image and identified the liquid droplets as having a circularity of 0.8–1.The results are reported for five replicates (**Supplementary file 2**). At 50 mM concentrations the droplet sizes were monitored for 16 min (**Supplementary file 3**).

### Determination of the droplet distributions of FUS

The previously published data on FUS droplets (**Kar et al., 2022**) were used, where the fluorescence intensity of the FUS droplets was reported to be proportional to their diameters (**Kar et al., 2022**). The abundance of untagged FUS droplets formed at 0.125, 0.25, 0.5, and 1.0, 2.0 μM concentrations were measured by nanoparticle tracking analysis for 51 different droplet sizes with three repetitions for each measure (**Kar et al., 2022**). Similarly, the abundance of SNAP-tagged FUS droplets at 0.125, 0.25, 0.5, 1.0, 1.5, 2.0, 2.5, and 3.0 μM concentrations were measured for 64 different sizes with 3 repetitions for each measure (**Kar et al., 2022**). For each data point, the mean droplet abundance and the standard error of the mean from the three experimental replicates were reported. The SDF was computed as $P_{>x} = N_{>x}/N_{tot}$, where $N_{>x}$ is the number of droplets above size $x$, and $N_{tot}$ is the number of droplets. The values of $x$ were chosen according to the original data (**Kar et al., 2022**).

### Determination of the critical exponents for FUS

We computed the $k$th moment of each droplet size, determined the average for each value of $k$ and computed the ratios of the subsequent moments (**Equation 4**). We tested several $k$ values from 0.25 to 3 in 0.25 increments, determined the moment ratios in each case. We selected four $k$ values ($k = 0.5$, $k = 1.0$, $k = 1.5$, and $k = 2.0$) where the moment ratios at each concentration provided the best linear fit for both FUS and SNAP-tagged FUS. Then we generated a log–log plot of the moments and $\tilde{\rho}$, the distance from the critical concentration (**Figure 1**). For each value of $k$, the points were estimated by taking the average of the computation done on the three independent measurements. The errors of the data points are estimated as the standard error of the mean. At each $k$ value, we performed a linear regression between the points at each concentration (**Kar et al., 2022**) determining the slope, resulting in four $\varphi_k$ values, with an associated error from the weighted linear regression. Given the independent observations $\varphi_k$ with variance $\sigma_k$, the value of the exponent was obtained as the weighted average

$$\bar{\varphi} = \frac{\Sigma_k \frac{\varphi_k}{\sigma_k^2}}{\Sigma_k \frac{1}{\sigma_k^2}} \tag{15}$$

The error on the exponent $\varphi$ was computed as

$$Var\left(\bar{\varphi}\right) = \frac{1}{\sum_k \frac{1}{\sigma_k^2}} \tag{16}$$

The exponent $\alpha$ was determined based on the log–log plot of the mean of the droplet size distribution at various distances from the critical concentration ($|\tilde{\rho}|$) (**Figure 1**). We performed a linear

regression weighted by the inverse of the variance of the data points corresponding, as previously, to the computed mean of the three independent experiments for each concentration (*Equation 6*) using $\varphi = 1$. The error on the slope from the weighted linear regression is associated with the estimate of the exponent $m$. The exponent $\alpha$ is then derived using $\alpha = 1 - m$.

At each concentration, from the distribution of ln(s) we determined $s_0$ as $<\ln(s)> = \ln(s_0)$, and $\sigma$ as the standard deviation of ln(s). At each concentration, we plotted the droplet size distribution function versus $ln\left(s/s_0\right)/\sigma$ (*Equation 10*) using the $s_0$ and $\sigma$ values determined from the corresponding droplet size distribution at the given concentration. The log-normal behaviour is demonstrated by the resulting collapse (*Figure 2*). Furthermore, the collapsed curves overlapped with a theoretical log-normal curve computed with $s_0 = 1$, $\sigma = 1$, which is the normal distribution in the rescaled log variables.

### Determination of the critical concentration for FUS

The $k$th moment of the log-normal distribution (*Equation 13*) and the scaling ansatz with $\alpha = 0$ (*Equation 3*) gives

$$\left(\left\langle s^k\right\rangle\right)^{\frac{-1}{k}} = \frac{1}{s_c}e^{\frac{(3-k)\sigma^2}{2}} \tag{17}$$

where $s_c = a \cdot \left|\frac{\rho-\rho_c}{\rho_c}\right|^{-1}$, $a$ is a proportionality constant independent of $k$, and $\sigma$ is the standard deviation of the logarithm of the droplet probability size distribution. Therefore,

$$\left(\left\langle s^k\right\rangle\right)^{-1/k} = \frac{1}{a}e^{\frac{(3-k)\sigma^2}{2}}\left|\frac{\rho-\rho_c}{\rho_c}\right| \tag{18}$$

plotted versus the concentration intersects the $x$-axis at the value of the critical concentration.

### Determination of the critical exponents for α-synuclein

For α-synuclein, the critical exponents were determined in two different ways, the first using the same method as for FUS (*Equation 18*), and the second using

$$\left(\frac{<s^k>}{<s>}\right)^{-1/(k-1)} = \left(\frac{c_1}{c_k}\right)^{1/(k-1)}\frac{1}{a}\left|\frac{\rho-\rho_c}{\rho}\right|^{\varphi} \tag{19}$$

### Determination of the critical concentration of α-synuclein

As for the critical exponents, we determined the critical concentration of α-synuclein using the scaling ansatz in two different ways, either using *Equation 18* or using *Equation 19* with the constrain $\varphi = 1$. We plotted the moments of the droplet size (*Equation 3*) versus the concentration, using a range of values of $k$ to cover majority of the data. Analogously to what happens in *Figure 3*, the lines plotted in *Figure 5D* intercept the $x$-axis on the same point, which corresponds to the critical concentration. We performed a linear regression weighted by the inverse variance, determining the intercept on the $x$-axis. For each value of $k$, we calculated the regression errors of both intercept and slopes. We estimated the critical concentration from each fit as the intercept calculated as $\rho_{c,i} = -q/m$, where $q$ and $m$ are the $y$-axis intercept and the slope retrieved by the fit, respectively, where the subscript $i$ indicates each independent experiment. The estimated $\rho_c$ was obtained as the mean of the different independent values $\rho_{c,i}$. The error on the estimate of $\rho_c$ is obtained as the standard error of the mean of the $\rho_{c,i}$.

## Acknowledgements

We acknowledge insightful discussions with Drs. Serena Carra, Jonathan Vinet, Alex Buell, and Soumik Ray. This work was supported by AIRC IG 26229 (MF).

# Additional information

## Funding

| Funder | Grant reference number | Author |
|--------|------------------------|--------|
| AIRC | IG 26229 | Monika Fuxreiter |

The funders had no role in study design, data collection, and interpretation, or the decision to submit the work for publication.

## Author contributions

Tommaso Amico, Data curation, Software, Formal analysis, Investigation, Visualization, Writing – review and editing; Samuel Toluwanimi Dada, Andrea Lazzari, Antonio Trovato, Data curation, Formal analysis, Validation, Investigation, Visualization, Methodology, Writing – review and editing; Michaela Brezinova, Data curation, Software, Formal analysis, Validation, Investigation, Visualization, Methodology, Writing – review and editing; Michele Vendruscolo, Monika Fuxreiter, Amos Maritan, Conceptualization, Data curation, Formal analysis, Supervision, Funding acquisition, Validation, Investigation, Visualization, Methodology, Writing – original draft, Project administration, Writing – review and editing

## Author ORCIDs

Samuel Toluwanimi Dada (iD) https://orcid.org/0000-0003-4816-9852
Andrea Lazzari (iD) https://orcid.org/0009-0000-3387-4378
Michele Vendruscolo (iD) https://orcid.org/0000-0002-3616-1610
Monika Fuxreiter (iD) https://orcid.org/0000-0002-4463-6727

Reviewer #1 (Public Review): https://doi.org/10.7554/eLife.94214.3.sa1
Reviewer #2 (Public Review): https://doi.org/10.7554/eLife.94214.3.sa2
Author response https://doi.org/10.7554/eLife.94214.3.sa3

---

# Additional files

## Supplementary files

• Supplementary file 1. Table S1. Size distributions of FUS and SNAP-FUS condensates below their critical concentrations.

• Supplementary file 2. Table S2. Size distributions of α-synuclein condensates below their critical concentration.

• Supplementary file 3. Table S. Size distributions of α-synuclein condensates at 50 µM concentration as a function of the time.

• MDAR checklist

## Data availability

All the data are provided in the manuscript and supplementary information.

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
