## [Editor Report · eLife assessment]

In this **valuable** study, the authors analyze droplet size distributions of multiple protein condensates and their fit to a scaling ansatz, highlighting that they exhibit features of first- and second-order phase transitions. The experimental evidence is **solid**, and it prompts further research into the nature of the link between percolation and phase separation.

---

## [Referee Report · Reviewer #1 (Public Review)]

The authors analyse droplet size distributions of multiple protein condensates and fit to a scaling ansatz to highlight that they exhibit features of first-order and second-order phase transitions. While the experimental evidence is solid, the text lacks connection and contextualization to the well-understood expectations from the coupling of percolation and phase separation in protein condensates - a phenomenon that is increasingly gaining consensus amongst the community. The evidence supports the percolatoin+phase separation model rather than being close to a true critical point in the liquid-gas phase space. Overall, the work is useful to the community.

Strengths:

The experimental analysis of distinct protein condensates is very well done and the reported exponents/scaling framework provides a clear framework to help the community help deconvolve signatures of percolation in condensates.

Weaknesses:

The principal concern this reviewer has is that the reviewers adopt a framing in this paper to present a discovery of second-order features and connections to criticality - however they ignore/miss the connections to percolation (a well-understood second-order transition that is expected to play a major role in protein condensates). I believe this needs to be addressed and the paper suitably revised to help connect with these expectations.

- Protein condensates have been increasingly understood to be described as fluids whose assembly is driven by a connection of density (phase separation, first-order) and connectivity (percolation, second-order) transitions. This has been long known in the polymer community (Flory, Stockmayer, Tanaka, Rubinstein, Semenov and others) and recently repopularized in the condensate community (by Pappu and Mittag, in particular, amongst others). The authors make no connections to any of this frameworks - which actually seem to be the essence of what they are describing.

- Percolation theory, which has been around for more than half-a-century, has clear-cut scaling laws that have essentially similar forms to the ansatz adopted by the authors and the commonalities/differences are not discussed by the authors - this is essential since this provides a physical basis for their ansatz rather than an arbitrary mathematical formulation. In particular, percolation models connect size distribution exponents to factors like dimensionality, valence, etc. and if these connections can be made with this data, that would be very powerful.

- The connections between spinodal decomposition and second-order phase transitions are very confusing. Spindal decomposition happens when the barriers for first-order phase transitions are zero and systems can phase separate without crossing nucleation barriers. Further, the "criticality" discussed in the paper is confusing since it more likely refers to a percolation threshold and much less likely to a "critical temperature" (Tc -where spinodal and binodals become identical). I would recommend reframing this argument.

It's unlikely, in this reviewer's opinion, that the authors are actually discussing a "first-order" liquid-gas critical point - because saturation concentrations of these proteins can be much higher with temperature and the critical point would thus likely be at much higher concentrations (and ofc temperature). Further the scaling exponents don't fall in that class naturally. However, if the authors disagree, I would appreciate clear quantitative reasons (including through the scaling exponents in that universality class) and be happy to be convinced to change my mind. As provided, the data does not support this model.

---

## [Referee Report · Reviewer #2 (Public Review)]

In response to the two referee reports, the authors have made substantial improvements. Regarding my previous concerns, the new data provided in Fig.6 for demonstrating that the droplet size distribution is stable over time is particularly valuable.

As to several of my other previous concerns regarding possible change in droplet size distribution over time, etc., the authors responded by stating that their system was below the critical concentration and therefore the possible scenarios pointed out in my previous report were not expected. While there may be a certain degree of validity to their argument, it would be much more helpful to the readers if the authors would bring up my previous concerns briefly (as readers of the journal will likely have similar concerns) and then address them succinctly within the manuscript.

Apparently, as a key element in the authors' response to the referees, the term "transition concentration" in the originally submitted manuscript is now changed to "critical concentration" (including in the title and abstract). But the two terms do not have identical meaning. A transition concentration is usually recognized as the saturation concentration at which phase separation or some other transition process commences at a given temperature. The transition concentration can be lower than the critical concentration, whereas the critical concentration is associated with the critical temperature, above (or below, depending on the temperature dependence of phase separation) which phase separation is not possible. It will be best if the authors can clarify their usage of transition concentration vs. critical concentration in the version of record of their manuscript.

---

## [Author Response]

The following is the authors’ response to the original reviews

**eLife assessment**:In this useful study, the authors analyze droplet size distributions of multiple protein condensates and their fit to a scaling ansatz, highlighting that they exhibit features of first- and second-order phase transitions. The experimental evidence is still incomplete as the measurements were apparently done only at one time point, neglecting the possibility that droplet size distribution can evolve with time. The text would benefit from a connection to and contextualization with the well-understood expectations from the coupling of percolation and phase separation in protein condensates - a phenomenon that is increasingly gaining consensus amongst the community and that emphasizes "liquid-gas" criticality.

We have now carried out new experiments at multiple time points to establish that the droplet size distributions are stationary below the critical concentration. We have also addressed the comments made by the reviewers about the nature of the phase transition.

Our analysis does not depend on a specific hypothesis on the nature of the phase transition, whether it be percolation or a gas-liquid critical transition. The scaling that we observed is an emergent property that is independent from the possible theoretical models used to describe the phase transition. In fact, our scaling analysis indicates that any theoretical model proposed for protein phase separation should predict the critical exponents that we reported.

**Reviewer #1**:The authors analyse droplet size distributions of multiple protein condensates and fit to a scaling ansatz to highlight that they exhibit features of first-order and second-order phase transitions. While the experimental evidence is solid, the text lacks connection and contextualization to the well-understood expectations from the coupling of percolation and phase separation in protein condensates - a phenomenon that is increasingly gaining consensus amongst the community. The evidence supports the percolation and phase separation model rather than being close to a true critical point in the liquid-gas phase space. Overall, the work is useful to the community.

We are grateful to the reviewer for these positive comments. We would like to emphasises that our contribution is not to propose a theoretical model, but rather to report a scaling behaviour in the experimentally measured droplet size distributions. The main implication of our work is that any theoretical model should predict the scaling exponents that we derived from the experimental measurements.

Strengths:The experimental analysis of distinct protein condensates is very well done and the reported exponents/scaling framework provides a clear framework to help the community deconvolve signatures of percolation in condensates.Weaknesses:The principal concern this reviewer has is that the reviewers adopt a framing in this paper to present a discovery of second-order features and connections to criticality - however, they ignore/miss the connections to percolation (a well-understood second-order transition that is expected to play a major role in protein condensates). I believe this needs to be addressed and the paper suitably revised to help connect with these expectations.

The scaling that we found is not characteristic standard percolation, since the exponents that we obtained (a=0 and f=1) are different from those of percolation (a=1.19 and f=2.21). This difference indicates that protein phase separation is not in the same universality class of standard percolation. Further studies will be required to understand whether theoretical models based on percolation could predict the observed critical exponents.

- Protein condensates have been increasingly understood to be described as fluids whose assembly is driven by a connection of density (phase separation, first-order) and connectivity (percolation, second-order) transitions. This has been long known in the polymer community (Flory, Stockmayer, Tanaka, Rubinstein, Semenov, and others) and recently repopularized in the condensate community (by Pappu and Mittag, in particular, amongst others). The authors make no connections to any of these frameworks - which actually seem to be the essence of what they are describing.

As mentioned above, our purpose was neither to support an existing theoretical model, nor to propose a new one. Rather, we have reported a scaling behaviour and scaling exponents not noted before. Further studies will be required to establish whether existing theoretical models could account for this scaling behaviour.

- Percolation theory, which has been around for more than half a century, has clear-cut scaling laws that have essentially similar forms to the ansatz adopted by the authors, and the commonalities/differences are not discussed by the authors - this is essential since this provides a physical basis for their ansatz rather than an arbitrary mathematical formulation. In particular, percolation models connect size distribution exponents to factors like dimensionality, valence, etc. and if these connections can be made with this data, that would be very powerful.

The scaling ansatz that we are using is commonly adopted in studies of critical phenomena, and it is not specific to percolation. The scaling exponents depends only on very few attributes like dimensionality, symmetries and if interactions are short or long range. These attributes determine the universality class. As such, scaling does not link with molecular determinants, but can distinguish different classes.

- The connections between spinodal decomposition and second-order phase transitions are very confusing. Spindal decomposition happens when the barriers for first-order phase transitions are zero and systems can phase separate without crossing nucleation barriers. Further, the "criticality" discussed in the paper is confusing since it more likely refers to a percolation threshold and much less likely to a "critical temperature" (Tc -where spinodal and binodals become identical). I would recommend reframing this argument.

We cannot refer to percolation threshold as our model is not readily compatible with it. We elaborated and better explained the differences between these models.

It's unlikely, in this reviewer's opinion, that the authors are actually discussing a "first-order" liquid-gas critical point - because saturation concentrations of these proteins can be much higher with temperature and the critical point would thus likely be at much higher concentrations (and ofc temperature). Further, the scaling exponents don't fall into that class naturally. However, if the authors disagree, I would appreciate clear quantitative reasons (including through the scaling exponents in that universality class) and be happy to be convinced to change my mind. As provided, the data does not support this model.

We have now clarified in the manuscript that we do not discuss the liquid-gas critical point.

**Reviewer #2**:This is a potentially interesting study addressing a possible scale-invariant log-normal characteristic of droplet size distribution in the phase separation behavior of biomolecular condensates. Some of the data presented are valuable and intriguing. However, as it stands, the validity and utility of this study are uncertain because there are serious deficiencies in the execution and presentation of the authors' results. Many of these shortcomings are fundamental, including a lack of clarity in the basic conceptual framework of the study, insufficient justification of the experimental setup, less-than-conclusive experimental evidence, and inadequate discussion of implications of the authors' findings to future experimental and theoretical studies of biomolecular condensates. Accordingly, this reviewer considers that the manuscript should undergo a major revision to address the following. In particular, the discussion should be significantly expanded by including references mentioned below as well as other references pertinent to the issues raised.

We thank the reviewer for the helpful comments. In the revised version of the manuscript we clarified that we aimed to use a well-established tool – the scaling analysis – to study phase transition and applied to the protein condensation process. This approach offers insight into a universal aspect of protein phase separation, and also provides a practical approach to determine the phase boundary. The observed fat-tailed distribution of protein droplet sizes is not what is normally observed in more standard phase separation systems in the subsaturated phase. Our contribution is not to propose a theoretical model, but rather to report the observation of a scaling behaviour.

(1) The theoretical analysis in this study is based on experimental data on condensed droplet size distributions for FUS and α-synuclein. The size data for FUS droplet is indirect as it relies on the assumption that FUS droplet diameter is proportional to fluorescence intensity of labeled FUS (page 10 of manuscript), with fluorescence data adopted from a previously published work by another group (Kar et al. & Pappu, ref.27). Because fluorescence of a droplet is expected to be dependent upon the condensed-phase concentration of FUS, this proportional relationship, even if it holds, must also be modulated by FUS concentration in the droplet. Moreover, why should fluorescence be proportional to diameter but not the cross-sectional area or volume of the FUS droplet, which would be more intuitive? These issues should be clarified. A new measure by microscopy is used to determine the size distribution of condensed α-synuclein; but no microscopy image is shown. It is of critical importance that such raw data (for example microscopy images) be presented for the completeness and reproducibility of the experiment because the entire study relies on the soundness of these experimental measurements.

As we mentioned in the article, for the scaling analysis, the droplet dimensions could be assessed in 1D (length), 2D (area) or 3D (volume). For the FUS experiments, we used the data as the authors provided in the original publication (PNAS 2022). For alpha-synuclein, we provided the data in the article.

(2) Despite the authors' claim of a universal scaling relationship, the log-log scatter plots in Figure 1 (page 15 of the manuscript) exhibit significant deviations from linearity at low protein concentrations (ρ→0). Given this fact, is universal scaling really valid? Discussion of this behavior is conspicuously absent (except the statement that these data points are excluded in the fit). In any case, the possible origins of these deviations should be thoroughly discussed so that the regime of universal scaling can be properly delineated.

In general, one would expect the scaling ansatz to be valid close to the phase boundary. It is the feature of the ansatz, that further away from the boundary, deviations are expected because of the decreasing relevance of critical phenomena.

(3) Droplet size distribution most likely depends on the time duration after the preparation of the sample. For α-synuclein, "liquid droplet size characterisation images were captured 10 minutes post-liquid droplet formation" (page 9 of the manuscript). Why 10 minutes? Have the authors tried imaging at different time points and, if so, do the distributions at different time points remain essentially the same? If they are different, what is the criterion for focusing only on a particular time point? Information related to these questions should be provided.

We have now determined the droplet size distribution of alpha-synuclein at different time points, finding that they are not dependent on time within experimental uncertainties (Figure 6 in the revised manuscript).

(4) At least two well-known mechanisms can lead to the time-dependent distribution of liquid droplet sizes: (i) coalescence of droplets in spatial proximity to form a larger droplet, and (ii) Ostwald ripening, i.e., formation of larger droplets concomitant with the dissolution of smaller droplets without fusion of droplets. The implications of these mechanisms on the authors' droplet size distributions should be addressed. Indeed, maintaining a size distribution against these mechanisms in vivo often requires active suppression [Bressloff, Phys Rev E 101, 042804 (2020)] with possible involvement of chemical reactions [Kirschbaum & Zwicker, J R Soc Interface 18, 20210255 (2021)]. These considerations are central to the basic rationale of this study and therefore should be carefully tackled.

These two mechanism of growth are relevant above the critical concentration. Below the critical concentration, which is the regime that we investigated in our work, there is no need of active suppression.

(5) If coalescence and/or Ostwald ripening do occur, given sufficient time after sample preparation, the condensed phase may become a single large "droplet" or a single liquid layer. Does this occur in the authors' experiments?

As we are below the critical concentration, this is unlikely to occur, as indeed supported by the experiments mentioned at point (3).

(6) It is unclear whether the authors aim to address the kinetic phenomenon of liquid droplet formation and evolution or equilibrium properties. The two types of phenomena appear to be conflated in the authors' narrative. Clarification is needed. If this work aims to address timeindependent (or infinite-time) equilibrium properties, how are they expected to be related to droplet size distribution, which most likely is time-dependent?

Our analysis focuses on the equilibrium properties of the droplet size distribution below the critical concentration, and it should guide the proposal of a theoretical model that explains the emergence of scaling. In the introductory part of our manuscript, we proposed a possible scenario that tries to extend the Flory-Huggins’s theory to predict a scaling behaviour appropriate to a critical transition. Other scenarios are possible, and our result along with further experiments are needed to arrive at a deeper understanding of protein aggregation.

(7) The relationship between the potentially time-dependent droplet size distribution and equilibrium properties of ρt and ρc (transition and critical concentrations, respectively) should be better spelled out. An added illustrative figure will be helpful.

We are addressing equilibrium properties, not kinetic ones. See also the answers to point 6.

(8) The authors comment that their findings appear to be inconsistent with Flory-Huggins theory because Flory-Huggins "characterizes droplet formation as a consequence of nucleation ..." (page 8 of the manuscript). Here, three issues need detailed clarification: (i) In what way does Flory-Huggins mandate nucleation? (ii) Why are the findings of apparent scale invariance inconsistent with nucleation? (iii) If liquid droplet formations do not arise from nucleation, what physical mechanism(s) is (are) envisioned by the authors to be underpinning the formation of condensed liquid droplets in protein phase separation?

We do agree that the Flory-Huggins theory does not mandate nucleation above the spinodal line. However, we are addressing the equilibrium properties below the critical concentration, so the stable phase is the dilute phase, and there is no nucleation.

(9) Are any of the authors' findings related to finite-system effects of phase separation [see, e.g., Nilsson & Irbäck, Phys Rev E 101, 022413 (2020)]?

Our experimental system is macroscopic, so we would not expect finite size effects.

(10) Since the authors are using their observation of an apparent scale-invariant droplet size distribution to evaluate phase separation theory, it is important to clarify whether their findings provide any constraint on the shape of coexistence curves (phase diagrams).

We are only reporting the phenomenological observation of a scaling behaviour, so we may not speculate at this stage on the constraints of the coexistence curves. This is indeed an exciting opportunity for future studies.

(11) More specifically, do the authors' findings suggest that the phase diagrams predicted by Flory-Huggins are invalid? Or, are they suggesting that even if the phase diagrams predicted by Flory-Huggins are empirically correct (if verified by experimental testing), they are underpinned by a free energy function different from that of Flory-Huggins? It is important to answer this question to clarify the implications of the authors' findings on equilibrium phase behaviors and the falsifiability of the implications.

As mentioned above, our main conclusion is that the droplet size distribution follows a scaling behaviour. Our contribution is not to propose a theoretical model, but rather to propose a scaling behaviour that should be accounted for by existing of future theoretical models.

(12) How about the implications of the authors' findings on other theories of protein phase separation that are based on interactions that are different from the short spatial range interactions treated by Flory-Huggins? For instance, it has been observed that whereas the Flory-Huggins-predicted phase diagrams always convex upward, phase diagrams for charged intrinsically disordered proteins with long spatial range Coulomb interactions exhibit a region that concave upward [Das et al., Phys Chem Chem Phys 20, 28558-28574 (2018)]. Can information be provided by the authors' findings regarding apparent scale-invariant droplet size distribution on the underlying interaction driving the protein molecules toward phase separation?

This is an interesting point for future studies about the type of interactions that give rise to the observed scaling behaviour.

(13) Table S1 (page 4) and Table S2 (page 7) are mentioned in the text but these tables are not in the submitted files.

We have added the Supplementary Tables as well as the source files for the figures.

(14) The two systems studied (FUS and α-synuclein) have a single intrinsically disordered protein (IDP) component. It is not clear if the authors expect their claimed scaling relation to be applicable to systems with multiple IDP components and if so, why.

From the data that we have currently analysed, we feel that we may not speculate on this interesting point, leaving it to future studies.